# Biological Extraction, HPLC Quantification and Medical Applications of Astaxanthin Extracted from Crawfish “*Procambarus clarkii*” Exoskeleton By-Product

**DOI:** 10.3390/biology11081215

**Published:** 2022-08-13

**Authors:** Salwa Hamdi, Nour Elsayed, Mohamed Algayar, Verina Ishak, Mariam Ahmed, Sara Ahmed, Mohamed Kamal, Mohamed Abd El-Ghany

**Affiliations:** 1Zoology Department, Faculty of Science, Cairo University, Giza 12613, Egypt; 2Biotechnology / Molecular Biochemistry Program, Faculty of Science, Cairo University, Giza 12613, Egypt; 3Botany and Microbiology Department, Faculty of Science, Cairo University, Giza 12613, Egypt

**Keywords:** astaxanthin, *Procamburus clarkii* waste, *Saccharomyces cerevisiae*, *Lactobacillus lactis*, *Bifidobacterium lactis*, *Candida utilis*

## Abstract

**Simple Summary:**

The study aims to provide an eco-friendly method for the extraction of the natural pigment called “astaxanthin” from crawfish powder. Bacterial and fungal strains that are beneficial to humans were used as an alternative method of extraction instead of using chemicals. Astaxanthin concentration was determined using an analytical tool referred to as “high-performance liquid chromatography” (HPLC). The results were promising; however, future studies can provide more effective methods for obtaining better results. Several tests were done to evaluate the biological activity of extracted astaxanthin such as antifungal, anti-inflammatory, antioxidant, and anticancer. Further purification of the extracted astaxanthin from crawfish exoskeleton needed to be done to assure that the results of the tests obtained were only due to the action of astaxanthin.

**Abstract:**

The main challenge of astaxanthin extraction is to provide an eco-friendly method of extraction instead of chemical methods that harm human health. This study provided an eco-friendly method for astaxanthin extraction using two bacterial and fungal probiotics (*Bifidobacterium lactis, Lactobacillus lactis, Candida utilis*, and *Saccharomyces cerevisiae*, respectively) and determined the astaxanthin concentration by high-performance liquid chromatography (HPLC) analysis. The results showed that the highest concentration was obtained by *S. cerevisiae* (45.69 µg/g). Several biological tests were done on the exoskeleton containing astaxanthin of crawfish. Antifungal activity was effective against *C. utilis* (inhibition zone is 12.3 ± 0.5 mm). The scavenging percentage of 2,2-diphenyl-1-picrylhydrazyl (DPPH scavenging percentage) was 72.1% at 1000 µg/mL concentration of exoskeleton containing astaxanthin. The Hemolysis inhibition percentage was 65% at the same concentration used previously. Furthermore, the IC50 value of human liver cancer cell line (HepG2), human hepatocellular carcinoma (HCT), and breast cancer cell line MCF-7 were 24 µg/mL, 11 µg/mL, and 9.5 µg/mL, respectively. The least cell viability percentage was 19% (using breast cancer cell line (MCF-7)) at 100 µg/mL of exoskeleton containing astaxanthin. Thus, using microorganisms can be an alternative and promising way of astaxanthin extraction. Furthermore, purification of extracted astaxanthin is essential for medical applications.

## 1. Introduction

Crustacean taxonomy is constantly evolving, and the classification structure used herein based on Martin and Davis [1] reveals five classes with subterranean representatives: *Branchiopoda*, *Remipedia*, *Maxillopoda*, *Ostracoda,* and *Malacostraca* [2]. Every year, about 6 million to 8 million tonnes of waste crab, shrimp, crawfish and lobster shells are produced globally, with Southeast Asia alone producing approximately 1.5 million tonnes. Waste shells are frequently dumped in landfills or the sea in developing countries. Disposal of these shells can also be expensive in developed countries, costing up to US$150 per tonne in Australia, according to Chen et al. [3]. It is worth noting that around 80% of the crawfish exoskeleton is discarded [4]. This exoskeleton is rich in valuable substrates with significant resource potential [5].

Carotenoids are known for their antioxidant and anticancer activity in humans [6]. Carotenoids have been proven in several investigations to behave as antioxidants by quenching singlet oxygen and free radicals [7]. Astaxanthin is a type of carotenoid that exists mainly in aquatic animals to give the unique red color of crustaceans. It is also used as a natural pigment, so this critical pigment can be extracted from algae, yeast, and crustacean by-products which is our target [8]. It is a typical feed supplement used to boost growth and offer antioxidant and immunological benefits in a variety of aquatic animals [9]. Biological ensilation of shrimp wastes using lactic acid bacteria not only recovers astaxanthin but also enhances the proteins and chitin content in the fermented wastes while preserving astaxanthin’s chemical structure and biomedical capabilities [10]. The primary source of astaxanthin for human consumption is *Haematococcus pluvialis* [11].

According to many studies conducted in this field discovered that this type of xanthophyll carotenoid can be used as a therapeutic agent for numerous diseases because it shows no toxicity or side effects. Therefore, it is an effective anti-tumor due to its ability to prevent cancer cell migration, anti-apoptosis, anti-proliferation, and general support of the immune system [12]. However, the mechanism through which astaxanthin shows the anti-cancer properties is unknown [13]. Astaxanthin also exhibits preclinical antitumor activity in numerous cancer models, both in vivo and in vitro [14]. Astaxanthin has anti-cancer efficacy in various cancers, including the mouth, the bladder, colon, leukemia, and hepatocellular carcinoma, in addition to its significant anti-oxidative properties [15].

Astaxanthin antibacterial properties have only been studied in a few types of research showing that astaxanthin in the form of a nanoemulsion can kill a variety of gram-positive and gram-negative bacterial species that have MIC values of 500–4000 g/mL. In an in vitro study of *Trypanosoma cruzi*, doses of astaxanthin 200–300 g/mL reduced the parasite’s vitality [16]. It has been established in animal trials of mice infected with *Helicobacter pylori* that astaxanthin affects the bacterial abundance in the stomach [17].

During the last two decades, extensive research has shown the mechanism by which persistent oxidative stress leads to chronic inflammation, which then mediates the majority of chronic illnesses, including neurodegeneration and skin damage [18]. Many anti-inflammatory pharmacotherapies now include antioxidants and medicines that limit the synthesis of inflammatory mediators. Antioxidants act as free-radical scavengers within cells, are the first line of defense against free radicals [19]. More crucially, investigations have shown that astaxanthin can suppress the generation of inflammatory mediators in human keratinocytes by preventing NF-B activation, suggesting that astaxanthin could be a promising new treatment option for inflammatory skin illnesses [20].

The determination of astaxanthin content using high-performance liquid chromatography (HPLC) has been widely used. Wade et al. [21] used samples collected from the epithelium and shell of lobsters (*Panulirus cygnus*) of two colors red and white depending on stages of sexual maturity. The results revealed that carotenoid content found in the epithelium is much higher than in the shell of both red and white lobster in and, total carotenoid content in the epithelium of red lobster is more than that of white lobster. Furthermore, the esterified astaxanthin content in the epithelium of red lobster is more than that of the epithelium of white. In addition, various studies determined the astaxanthin concentration of shrimp by using HPLC where, the mobile phase was methanol: water: dichloromethane (DCM): acetonitrile which provided accuracy, precision, and reduction in sample preparation time [22].

The study’s major goal is to extract astaxanthin from crawfish exoskeleton by-product by a biological method using novel bacterial and fungal probiotics for the extraction, and quantification of extracted astaxanthin by HPLC analysis, and to study astaxanthin applications of antimicrobial, antioxidant, anti-inflammatory, and anticancer activity.

## 2. Materials and Methods

### 2.1. Crawfish and Raw Material Collection

Crawfish (*Procambarus clarkii*) were collected and delivered on ice from Rashid or Rosetta, The Nile Delta’s port city


**Raw material:**


Bacteria (*Bifidobacterium lactis*–accession no: DSM10140), (*Lactobacillus lactis*-ACA-DC 178-accession number LS991409), and fungus (*Candida utilis*–accession no: NRRL Y-660), (*Saccharomyces cerevisiae*- accession no: 006-001) were presented by the Microbiological Resources Centre (Cairo Mircen), Faculty of Agriculture, Ain Shams University. Egypt. Human hepatocellular carcinoma (HCT), breast cancer cell line (Michigan Cancer Foundation-7) (MCF-7), and human liver cancer cell line (HepG2) were purchased from the American type culture collection (USA)

### 2.2. Sample Processing

Crawfish frozen samples were left at room temperature. Later carapace, shells, and legs were separated from the samples. All the waste products were rinsed with fresh water then all the shells were weighed before drying [23]. The shells were then dried for 10 h in a 50 °C oven. After that, it was allowed to air dry for another 24 h. The shells were then crushed to a fine powder with a home blinder and sorted through a sieve to obtain a fine powder, which was then weighed and stored at room temperature in clean containers with silica packets.

### 2.3. Biological Method for Astaxanthin Extraction

#### 2.3.1. Microorganisms and Culture Media

Subculture of two bacterial probiotics which are *B. lactis* (DSM10140) and *L. lactis* on MRS Agar medium [meat extract 8.0 g, yeast extract 4.0 g (MERCK), MnSO_4_ 0.04 g, MgSO_4_ 0.2 g, peptone from casein 10.0 g, C_6_H_14_N_2_O_7_ 2.0 g, CH_3_COONa 5.0 g, D(+)glucose 20.0 g, tween 80 1.0 g], then solidifying the liquid medium by adding 15 g/L and left for 48–72 h incubation in the presence of 5% CO_2_ [24]. However, the solidification step in our method was not performed.

Czapek-Dox agar medium was used for the cultivation of two fungal probiotics *C. utilis* and *S. cerevisiae* where the medium composition is [K_2_HPO_4_ 1 g, sucrose 20 g, KCl 0.5 g, FeSO_4_.7H_2_O 0.01 g, NaNO_3_ 2 g, MgSO_4_.7H_2_O 0.5 g, agar 15 g] [25]. The media was incubated for 72 h at 37 °C, where the solidification step in our method was excluded

The two bacterial probiotics and *C. utilis* are approved for human consumption by FDA [26,27] in addition, *S. cerevisiae* is classified as biosafety level 1 [28]. Thus, no biosecurity-related problems will occur as long as the standard microbiological practices are followed. 

#### 2.3.2. Extraction Steps

After preparing the liquid MRS and Czapek-Dox media (both have an equal volume of 100 mL) 10 g of powder were added to each media, and then all the flasks containing the liquid media and powder were autoclaved. Moreover, 1mL of each bacterium and fungi was inoculated to the corresponding media after autoclaving. Then, all the prepared media containing probiotics were incubated for 7 days at 100 rpm.

### 2.4. Lyophilization

The cooling lyophilization was applied to each sample using “Edwards modulyo freeze dryer” at −45 °C and 10 atm for 48 h. 

### 2.5. HPLC Analysis:

This analysis method was done according to Lu et al. [29] at National Research Centre, Egypt.

#### 2.5.1. HPLC Conditions

HPLC analysis was carried out using an Agilent 1260 series. The separation was carried out using the Eclipse C18 column (4.6 mm × 250 mm i.d., 5 μm). The mobile phase consisted of methanol: water: DCM: acetonitrile (70:4:13:13) (*v*/*v*) at a flow rate of 1 mL/min. The mobile phase was consecutively programmed in an isocratic system. The diode array detector was monitored at 280 nm. The injection volume was 5 μL for each of the sample solutions. The column temperature was maintained at 40 °C.

#### 2.5.2. Sample Preparation

A total of 15–20 mg of extract was dissolved in 1 mL acetone and vortex for 1 min, sonicated for 15 min, and then filtered by a 0.45-micron filter.

### 2.6. Antimicrobial Activity Assay 

The Antimicrobial activity was performed using the modified diffusion disk method [30]. In this method, 100 µL of fungal or bacterial probiotics were cultivated in fresh media with 10 mL to approximately reach 108 cells/mL or 105 cells/mL according to bacterial or fungal probiotics [31]. In agar plates, 100 µL of probiotics suspension was spread and incubated for 48 h at 25 °C regarding *Aspergillus flavus* as a type of fungi whose positive control was Nystatin**,** and for both types of bacteria Gram-positive which includes (*Bacillus subtilis*- *Streptococcus faecalis*- *Staphylococcus aureus*) where ampicillin was a positive control and Gram-negative which includes (*Neisseria gonorrhoeae- Escherichia coli*- *Pseudomonas aeruginosa*) where gentamycin was a positive control. Both were incubated for 24–48 h at 35–37 °C. Then, *Candida albicans* as yeast was incubated for 24–48 h at 30 °C. For negative control, 10 µL of distilled water, chloroform, and DMSO as a solvent was used to impregnate filter discs. The stock solutions tested concentrations consisting of methylene blue with 5 μg/mL plus glucose with 0.4 g/mL were used with 10µL to impregnate disks of blank paper with 8.0 mm diameter. A filter disk paper was immersed in the crawfish exoskeleton containing astaxanthin for testing which was added to the agar then the disk was diffused, and the chemical was placed around the disk. The inhibition zone appeared, and the diameters of inhibition zones were measured. This method was faster and simpler than broth methods [32].

### 2.7. Antioxidant Activity Assay

The crawfish exoskeleton containing astaxanthin antioxidant activity was evaluated against 2,2-diphenyl-1-picrylhydrazyl assay (DPPH) according to Chintong et al.; Navarro-Hoyos et al. [33,34]. A total of 100 µL of different astaxanthin concentrations dissolved in methanol were so added to 100 µL of DPPH dissolved in methanol. All samples were shaken well, left in the dark at room temperature for 30 min, and then measured at 517 nm using a spectrophotometer. Later a curve was driven for astaxanthin concentration against the % DPPHto get the astaxanthin-containing exoskeleton concentration required to reduce the initial DPPH content by half (EC_50_) and the DPPH inhibition scavenging effect was calculated as a percentage following this equation:%DPPH radical scavenging = [A_c_ − (A_s_ – A_sb_)/A_c_] × 100
where A_c_ the control sample is the absorbance of DPPH with methanol, A_s_ is the absorbance of the sample with DPPH and A_sb_ is the absorbance of the blank sample of astaxanthin with no DPPH.

### 2.8. Anti-Inflammatory Assay 

#### 2.8.1. Preparation of Erythrocyte Suspension:

Plasma was obtained by putting 3 mL of fresh whole blood into heparin tubes and then centrifugated at 3000 rpm for 10 min after. The sample was treated with an equal volume of saline to dissolve red blood pellets whose volume was measured to form a suspension with isotonic buffer solution (10 mM sodium phosphate buffer). The concentration of suspension was (40% *v*/*v*) and the composition of the buffer is 0.2 g monosodium phosphate, 1.15 g disodium phosphate and 9 g of sodium chloride, where all the constituents were dissolved in 1 L of distilled water [35].

#### 2.8.2. Hypotonicity Induced Haemolysis:

Erythrocyte suspension (0.1 mL) was added to three groups of tubes which were the control group (5 mL distilled water and 5 mL of indomethacin (standard) with a concentration of 200 µg/mL). The hypotonic solution (containing 5 mL of distilled water and gradual doses of extract, where the extract was either a standard or powdered sample of astaxanthin extracted by *S. cerevisiae* and doses were 100, 200, 300, 400, 500, 600, 800 and 1000 µg/mL). Furthermore, each dose was represented in duplicates. The isotonic solution contained sodium phosphate buffer and gradual doses of extract as explained in hypotonic solution). After adding the suspension to the samples, all mixtures were incubated at 37 °C for 1 h, followed by centrifugation at 1300 g for 3 min where the optical density of hemoglobin content was measured at 540 nm using a spectronic spectrophotometer. The percentage of haemolysis was calculated by considering the haemolysis that occurred in the control sample to be 100%, thus the percentage of inhibition of haemolysis was stated as [35]:$$\mathrm{percentage}\;\mathrm{of}\;\mathrm{haemolysis}\;\mathrm{inhibition}=\left[1-\frac{\mathrm{OD}2-\mathrm{OD}1}{\mathrm{OD}3-\mathrm{OD}1}\right]\times 100$$

OD_1_ = Absorbance of the test sample in isotonic solutionOD_2_ = Absorbance of the test sample in hypotonic solutionOD_3_ = Absorbance of the control sample in hypotonic solution

### 2.9. Anticancer Activity Assay

Human hepatocellular carcinoma (HCT), breast cancer cell line (Michigan Cancer Foundation-7) (MCF-7), and human liver cancer cell line (HepG2) were used to test the extract’s antitumor activities. The cells were cultured in RPMI-1640 media containing 10% inactivated fetal calf serum and 50 g/mL gentamycin. According to Ahmed et al. [36], the cells were sub-cultured two to three times a week at 37 °C in a humidified 5 percent CO_2_ environment. This analysis was done in the micro analytical center at Cairo university, Cairo, Egypt.

## 3. Statistical Analysis

All data presented in each experiment were means of triplicate assays. The SPSS 25 software was used in the determination of standard error (SE) (*p* < 0.05).

## 4. Results

### 4.1. HPLC Analysis

As shown in Figure 1, the astaxanthin concentration in the biological method using bacterial probiotics was higher using *L. lactis* (40.17 µg/g) than in *B. lactis* (39.48 µg/g) while using the fungal probiotics, astaxanthin concentration extracted using *S. cerevisiae* (45.69 µg/g) was higher than that using *C. utilis* (28.71 µg/g). In general, the concentration of extracted astaxanthin increased using *C. utilis*, *B. lactis*, *L. lactis*, and *S. cerevisiae,* respectively. While Figure 2 and Figure 3 show HPLC chromatogram for astaxanthin standard solution and HPLC chromatogram of astaxanthin extracted by the biological method using bacterial (*B. lactis, L. lactis*) and fungal probiotics (*S. cerevisiae* and *C. utilis*), respectively.

### 4.2. Antimicrobial Activity

As shown in Table 1, an inhibition zone was formed when using a crawfish exoskeleton containing astaxanthin against *Candida albicans* (the inhibition zone is 12.3 ± 0.5 mm). However, no inhibition zone was formed in the case of the gram-positive and the gram-negative bacteria used.

### 4.3. Antioxidant Activity

The crawfish exoskeleton containing astaxanthin extracted by *S. cerevisiae* shows that when the concentration of the sample is increased, the DPPH Scavenging rises as shown in Figure 4. The highest observed concentration was 1000 µg/mL, with a DPPH of 72.1%. The EC_50_ value of the sample is 131 µg/mL.

### 4.4. Anti-Inflammatory Activity

The percentage of haemolysis inhibition increases as the concentration increases, as shown in Figure 5, suggesting that the biological extract of the crawfish exoskeleton containing astaxanthin may have the potential to show anti-inflammatory activity—and the highest percentage of haemolysis inhibition is 65% at 1000 µg/mL sample.

### 4.5. Anticancer Activity

As shown in Figure 6, cell viability decreases by the gradual increase of the concentration of the crawfish exoskeleton containing astaxanthin where the IC_50_ value of HepG2, HCT, and MCF-7 tumor cell lines were 24 µg/mL, 11 µg/mL, and 9.5 µg/mL, respectively.

## 5. Discussion

A synthetic astaxanthin racemic form was approved by the US Food and Drug Administration (FDA) as an animal food additive after an extensive study of astaxanthin safety. The esterified astaxanthin compounds extracted from shrimp shells and natural microalgae were so approved [37]. Astaxanthin is increasingly being used as a dietary supplement in nutraceuticals, pharmaceuticals, and foods [38]. Astaxanthin provides a wide range of biological and physiological effects. It can be used to prevent various diseases such as intestinal damage [39] and in commercial uses. Due to its importance, there have been significant efforts to increase astaxanthin synthesis using biological sources rather than synthetic ones [40]

The HPLC analysis demonstrated that the highest concentration of extracted astaxanthin was found per our results in the biological method using *S. cerevisiae* (45.69 µg/g). Our results, when compared with the most common chemical methods, were higher than when ethanol was utilized by Yoon et al. [41], where (17.8 µg/g) of astaxanthin was extracted. According to Radzali et al. [42], the amount of extracted astaxanthin was (58.03 µg/g) when ethanol was utilized. In addition, Hu et al. stated that the concentration of astaxanthin extracted from shrimp (*procambarus clarkia*) using ethanol as solvent was (239.96 µg/g) [23].

In this study, the antimicrobial activity of astaxanthin extracted by *S. cerevisiae* was studied on a wide range of Gram-positive, Gram-negative bacteria and fungal strains, and our sample was effective against *Candida albicans* (ATTC: 10231). According to Santos et al. [43], the shrimp residue extract had little antimicrobial activity against *Staphylococcus aureus.* Dalei et al. [44] observed that the astaxanthin extracted by methanol has the highest antimicrobial activity against *Escherichia coli, Bacillus*, *Pseudomonas*, and *Staphylococcus.* While Akyön [45] reported that astaxanthin has antimicrobial activity against *Helicobacter pylori.*

Astaxanthin has an intense antioxidant activity due to its unique structure that includes keto (C=O) and hydroxyl (OH) endings, giving the astaxanthin the ability to donate hydrogen. As shown in Figure 4, The DPPH scavenging activity percentage was increased as the concentration of astaxanthin increased, and the lower EC_50_ value indicated higher antioxidant activity. The IC_50_ value of astaxanthin extracted using *S. cerevisiae* was 131 µg/mL which was much lower than the IC_50_ of the lobster (6675.25 µg/mL) as reported in Ngginak et al. [46]. However, our results were higher than the highest IC_50_ value of the astaxanthin extracted from *Haematococcus pluvialis* reported by zhang et al. [47], which equals 48.69 µg/mL.

The result of haemolysis inhibition suggests that crawfish containing astaxanthin treated with *S. cerevisiae* may have the potential to show anti-inflammatory activity. The cell viability in the anticancer test performed by Parvathy et al. [48] using astaxanthin and zinc oxide against MCF-7 Breast cancer cells, was 60 % which is much higher than the same concentration of crawfish exoskeleton containing astaxanthin treated with *S. cerevisiae* against the same cell line (19%) where the same study revealed that the cell viability reaches (18–20%) by using astaxanthin with cerium oxide nano-particles. Furthermore, the cell viability using astaxanthin extracted from *Haematococcus pluvialis* against the hepatocellular cell line was 50% (using 123.01 ± 2.97 µg/mL) [36] which is higher than the cell viability using crawfish exoskeletons containing astaxanthin treated with *S. cerevisiae* (24% as cell viability) (using 100 µg/mL as a concentration).

## 6. Conclusions

To conclude, extensive studies are needed to be done for extracting astaxanthin from different sources using different microorganisms. However, using microorganisms that are beneficial to humans is highly recommended. Conceptualizing that the astaxanthin extracted in an eco-friendly method has a significant biological activity and can be used in medical applications is an essential and promising step. The appropriate way to provide sufficient evidence for that conceptualizing is the purification of the exoskeleton of crawfish containing astaxanthin or any source of astaxanthin used.

## Figures and Tables

**Figure 1 biology-11-01215-f001:**
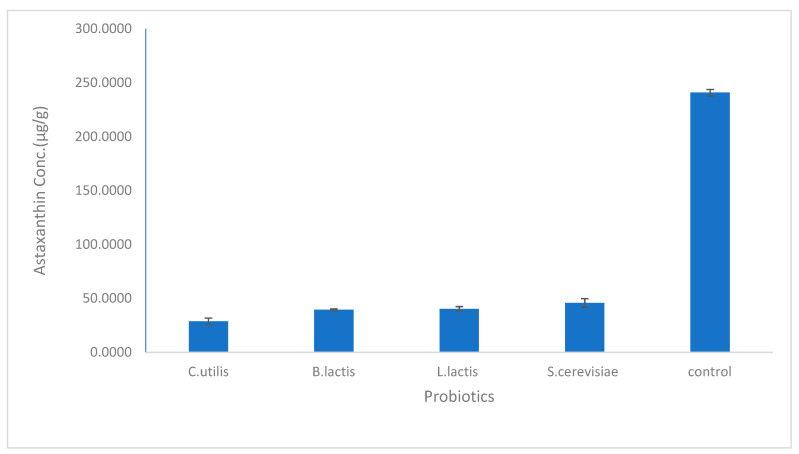
The concentration of astaxanthin extracted by bacterial and fungal probiotics (*Lactobacillus lactis- Bifidobacterium lactis- Saccharomyces cerevisiae- Candida utilis*). The mean ± SD (*p* <0.05).

**Figure 2 biology-11-01215-f002:**
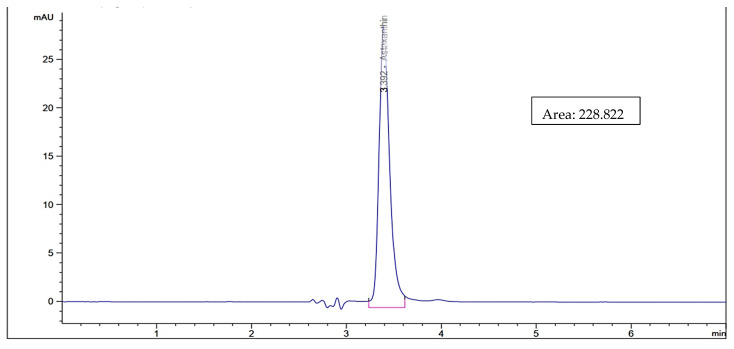
HPLC chromatogram for astaxanthin standard solution.

**Figure 3 biology-11-01215-f003:**
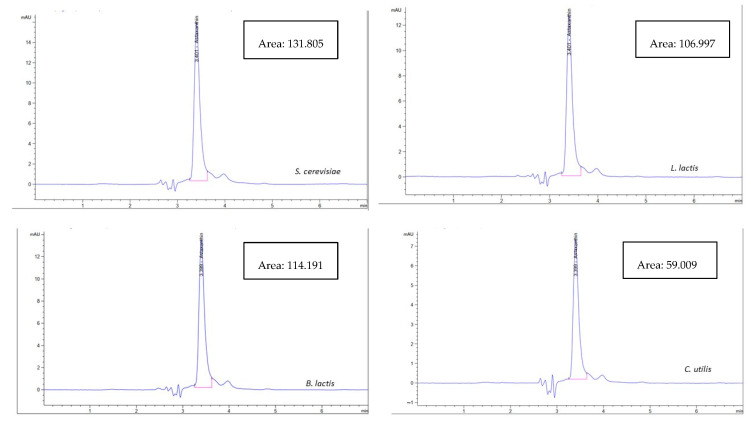
HPLC chromatogram of astaxanthin extracted by the biological method using bacterial (*B. lactis, L. lactis*) and fungal probiotics (*S. cerevisiae* and *C. utilis*).

**Figure 4 biology-11-01215-f004:**
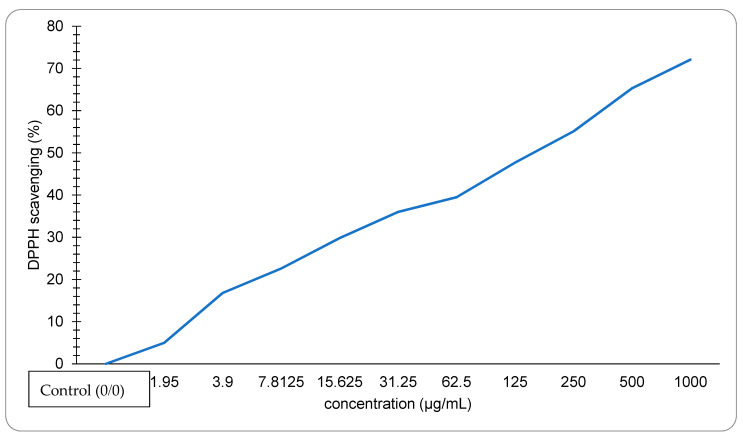
DPPH scavenging activity of astaxanthin extracted by *S. cerevisiae.*

**Figure 5 biology-11-01215-f005:**
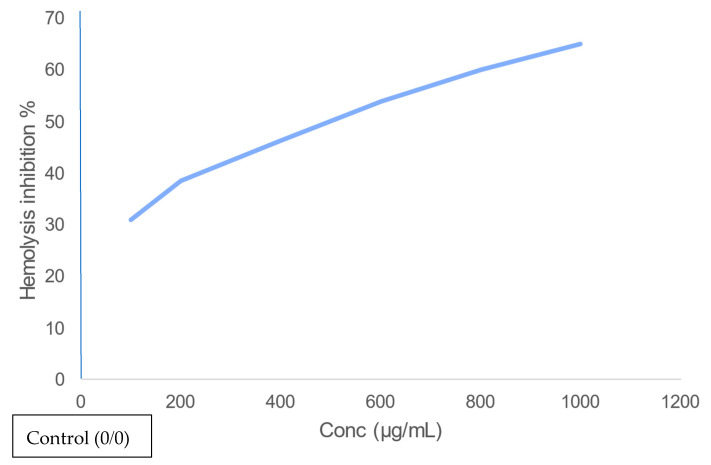
Anti-inflammatory activity of astaxanthin extracted using *S. cerevisiae* on erythrocytes.

**Figure 6 biology-11-01215-f006:**
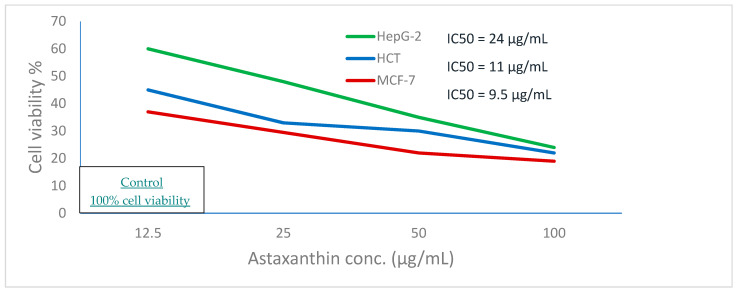
Anticancer activity of astaxanthin extracted from crawfish by S. cerevisiae against hepatocellular carcinoma (HepG-2), colon cancer (HCT), and breast cancer (MCF-7).

**Table 1 biology-11-01215-t001:** The antimicrobial activity of astaxanthin extracted by the biological method using *S. cerevisiae.*

Microorganism	Inhibition Zone Diameter (mm)		
	Sample	Control	
		Positive Control	Negative Control
**Gram positive**		Ampicillin	
** *Staphylococcus aureus* ** **(ATCC:13565)**	NA	20 ± 0.1	0.0
** *Staphylococcus aureus* ** **(ATCC:25923)**	NA	20 ± 0.1	0.0
** *Streptococcus mutans* ** **(ATCC:25175)**	NA	29 ± 0.5	0.1
**Gram negative**		Gentamycin	
** *Escherichia coli* ** **(ATCC:10536)**	NA	27 ± 0.5	0.0
** *Escherichia coli* ** **(ATCC:25955)**	NA	27 ± 0.5	0.0
** *Klebsiella pneumonia* ** **(ATCC:10031)**	NA	25 ± 0.5	0.0
** *Pseudomonas aeruginosa* ** **(ATCC:27853)**	NA	28 ± 0.3	0.0
**Fungal strains**		Nystatin	
** *Candida albicans* ** **(ATCC:10231)**	12.3 ± 0.5	21 ± 0.5	0.0
** *Aspergillus ochraceous* ** **(ATCC:22947)**	NA	18 ± 0.5	0.0
** *Aspergillus parasiticus* ** **(ATCC:26690)**	NA	18 ± 0.5	0.1
** *Aspergillus niger* ** **(ATCC:16404)**	NA	19 ± 0.5	0.0
** *Aspergillus niger* ** **(ATCC: MH368137)**	NA	19 ± 0.5	0.0

Negative control: DMSO; NA = no activity.

## Data Availability

The authors confirm that the data supporting the findings of this study are available within the article.

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
