# Peer review of "Biological Extraction, HPLC Quantification and Medical Applications of Astaxanthin Extracted from Crawfish “Procambarus clarkii” Exoskeleton By-Product"

_biology, 2022, doi:10.3390/biology11081215_

Round 1

Reviewer 1 Report

Improvement details

Simple summary: Improve redaction

Line 18: Missing point before “Thus….

Line 29, 36,252,255, 316: Enter micrograms correctly (ug) change by µg

Line 29: Before “Antifungal…., is missing a point

Line 30 and 300: (Inhibition zone is 12.3±0.5) is missing units % or mm ¿?, in addition, it must also indicate in the methodology, based on what scale an effective inhibition is considered. The positive controls gentamicin and nystatin are not mentioned in the methodology only in the results

Line 36: double parentheses

Line 36, 250, 251, 316, 386, 386, 387: Edit ml by mL

Line 36 to 38: Improve redaction

Line 39 to 40: Use italics in scientific names

Line 53 to 58: This part must be edited, to give continuity to the text

Line 194: Use the correct symbol ºC

Line 213, 217: change “ °c” by  “°C”

Line 220: Missing information on concentration,  “The stock solutions tested concentrations with 10 µ”

Line 224: word “disc”, must be “disk”

Line 351: HAET?

 Comments

1.- The introduction, is challenging to follow, and its content needs to be structured. The information is there, however, its content must be ordered.

 2.- The information generated by the authors is of valuable importance, however, it presents a deficiency in the writing, in addition to the lack of editing throughout the document, the figures can be improved. It is difficult to follow the methodological development, since methodologies such as the anticancer effect, which I consider relevant (the method used, is not developed, nor the controls used) in addition to a poor discussion of its results.

3.-With the hemolysis method, it cannot be concluded that astaxanthin has anti-inflammatory activity, it is also necessary to justify using this methodology. In terms of discussion of results, it is deficient.

4.-On the other hand, in what sense does it conclude that crawfish astaxanthin is more effective than shrimp astaxanthin, for this it would have to evaluate both matrices under the same conditions.

Author Response

All comments have been made in manuscript 

Reviewer 2 Report

In this study, bacteria and fungi were used to extract astaxanthin from crayfish powder, and high performance liquid chromatography (HPLC) was used to quantify and perform antifungal, anti-inflammatory, antioxidant, anti-cancer and other tests, but the experimental data was not comprehensive enough, and the article format was not standardized enough. The specific comments are as follows:

1. Some of the charts are not standardized. Fig.2 and Fig.4

2.The improper use of logical symbols in sentences . Eg.line69-71

3.There is no support for medical applications, and the application products in the medical field need to have a higher level of purification method .

4.The astaxanthin product activity verification index obtained is too small, whether it can increase the relevant indicators in the biological field, increase the logical persuasiveness, and present the research innovation point .

5.Whether the biological method of extracting astaxanthin will bring biosecurity-related problems, which needs to be further verified.

Author Response

All comments have been made 

Round 2

Reviewer 1 Report

Dear author

It still requires editing, I suggest you read carefully what you have written, in the part where you compare with other studies and in some other parts of the text. Improve figures.

Line 41: Change, ug/ml to µg/mL

Line 46, 354: Check parenthesis

Line 46-48: Edit, “Thus, the extraction of astaxanthin using microorganisms and purification of exoskeleton containing astaxanthin to evaluate astaxanthin biological activity accurately is highly recommended”

Line 100, 102, 104, 113: change the word Astaxanthin to astaxanthin

Line 123, 124, 296, 302: Change, Methanol by methanol; Dichloromethane by dichloromethane and Acetonitrile by acetonitrile. Acetone by acetone

Line 260: Before the word “After” must be a punctuation sign (.)

Line 280: punctuation sign (.)

Line 282, 284, 297, 299, 302, 306, 307, 308, 315, 318, 319, 370: Change, ml by mL, or µl by µL (please check all the text)

Line 288-289: Change, “Edwards Freeze Dryer Modulyo” by Edwards modulyo freeze dryer”

Line 298: change, Diode by diode

Line 303: change, 15min by 15 min

Line 319: “with 10 µ”, with 10 µL

Line 320: Immersed (immersed) in a chemical?, (be more specific)

Line 325-326: This paragraph needs editing

Line 353: eliminate Fig 5.

Line  357: Homologate hrs or h

Line 383-384: This paragraph needs editing (verb, and space between words)

Line 388: scientific names in cursive

Line 399-400: This paragraph needs editing

Edit figures 4, 5, 6 and 7

Line 424: is HepG2 not Hep-2

Line 431: is HepG2 not Hepg-2

Line 462: punctuation sign (.) before “However”

Author Response

All comments have been answered

Reviewer 2 Report

In this study, astaxanthin was extracted from crayfish powder using bacteria and fungi, and quantified by high performance liquid chromatography for antifungal, anti-inflammatory, antioxidant, and anti-cancer tests. After revision, the article is more complete, but the format of the article is not standardized enough. Specific comments and questions are as follows.

1The subheadings in the second part "Materials and Methods" and the third part "Results" have wrong serial numbers and need to be revised.

2Is there a control test in the hemolysis inhibition experiment?

3The format of the hemolysis formula needs to be revised.

4The diagram in Figure 1 is not standardized.

5Table 1, "Inhibition zone diameter (cm)", the unit "cm" is wrong, need to be modified.

6The diagram in Figure 4 is not standardized.

7The diagram in Figure 5 is not standardized.

8, Figure 6, 7 and 8 can be considered to be combined in one figure, more convenient to watch and read.

Author Response

All comments have been answered
